# Sustainable Preparation of Nanoporous Carbons via Dry Ball Milling: Electrochemical Studies Using Nanocarbon Composite Electrodes and a Deep Eutectic Solvent as Electrolyte

**DOI:** 10.3390/nano11123258

**Published:** 2021-11-30

**Authors:** Ana T. S. C. Brandão, Renata Costa, A. Fernando Silva, Carlos M. Pereira

**Affiliations:** Departamento de Química e Bioquímica, Faculdade de Ciências da Universidade do Porto, CIQUP–Physical Analytical Chemistry and Electrochemistry Group, Rua do Campo Alegre, s/n, 4169−007 Porto, Portugal; up200706627@edu.fc.up.pt (A.T.S.C.B.); renata.costa@fc.up.pt (R.C.); afssilva@fc.up.pt (A.F.S.)

**Keywords:** graphite, graphene, ball−milling tools, deep eutectic solvent, specific surface area, capacitance, electrical double−layer capacitor

## Abstract

The urgent need to reduce the consumption of fossil fuels drives the demand for renewable energy and has been attracting the interest of the scientific community to develop materials with improved energy storage properties. We propose a sustainable route to produce nanoporous carbon materials with a high−surface area from commercial graphite using a dry ball−milling procedure through a systematic study of the effects of dry ball−milling conditions on the properties of the modified carbons. The microstructure and morphology of the dry ball−milled graphite/carbon composites are characterized by BET (Brunauer–Emmett–Teller) analysis, SEM (scanning electron microscopy), ATR−FTIR (attenuated total reflectance–Fourier transform infrared spectroscopy) and Raman spectroscopy. As both the electrode and electrolyte play a significant role in any electrochemical energy storage device, the gravimetric capacitance was measured for ball−milled material/glassy carbon (GC) composite electrodes in contact with a deep eutectic solvent (DES) containing choline chloride and ethylene glycol as hydrogen bond donor (HBD) in a 1:2 molar ratio. Electrochemical stability was tracked by measuring charge/discharge curves. Carbons with different specific surface areas were tested and the relationship between the calculated capacitance and the surface treatment method was established. A five−fold increase in gravimetric capacitance, 25.27 F·g^−1^ (G40) against 5.45 F·g^−1^, was found for commercial graphene in contact with DES. Optimal milling time to achieve a higher surface area was also established.

## 1. Introduction

Carbon nanomaterials such as graphene and graphite present unique physical, chemical and mechanical properties [1]. These carbon nanomaterials have attracted much interest for a large diversity of applications, from energy conversion (e.g., solar and fuel cells) [2,3,4] to energy storage (e.g., supercapacitors and batteries) [5,6,7] and remediation (e.g., removal of heavy metals from water and soils) [8,9].

When applied to energy storage, carbon and its derivatives are excellent candidates as electrode materials due to physicochemical properties such as a low atomic number, making it a lightweight material with long term stability, low residual current, and broad potential range. From an economic point of view, carbon is also an excellent candidate due to its abundance, with the existence of both natural and synthetic sources [1] offering many choices for porous electrode construction. Parameters such as material dimensionality, specific surface area, pore volume, pore size distribution, particle size, and texture can be easily tuned by controlling the synthetic route parameters [10].

Many different carbon materials have been prepared and tested for supercapacitor applications during the past 20 years. According to a literature survey, within this period, new methods have been introduced to prepare high−capacitance materials at the laboratory scale, according to several peer reviews [1,11,12,13,14,15,16]. The most commonly used type of carbon for capacitive technologies is activated carbon, being recognized as a disordered agglomeration of nanoscale units made of graphene layers randomly oriented and strongly cross−linked, presenting a high surface area [1]; however, the production of activated carbon requires a high amount of energy [17].

Graphene is an atomically thin two−dimensional sp^2^ bonded carbon sheet, exhibiting remarkable properties such as excellent thermal optical and electrical conductivity, mechanical strength, and remarkable electrochemical properties making it a promising active material for supercapacitors [18]. Graphene−based supercapacitors were reported by Vivekchand et al. [19] with a specific capacitance of 75 F·g^−^^1^ and an energy density of 31.9 W·h·kg^−^^1^, in ionic liquids, in a two−electrode system with an electrode mass of 5 mg, with a current density of 1 A·g^−^^1^ and charge−discharge under the interval 0–1 V. The BET surface area of the material was around 925 m^2^·g^−^^1^, with an average pore size of 3 nm. This technique is rather important for the characterization of carbon materials. Raman spectroscopy is also a commonly used method for carbon−based material characterization, offering information regarding chemical modification, crystallite size and crystal disorder [20], and its resolution has been evolving (super−resolution), with the imaging time being reduced considerably [21,22,23].

The routes to obtain graphene/graphene oxide/graphite oxide require high costs and are therefore not sustainable for industrial production [24]. Nowadays, the synthesis of these carbon materials can be made using environmentally−friendly methods, with the use of nontoxic chemicals [25].

Another possible route to synthesize graphitic materials is ball milling, known to be an effective and ecological way to prepare high surface area materials compared to other methods such as chemical exfoliation [26] and allowing the reduction of particle size and the refinement of grains to a size below 1 µm. The strong forces created between the high−speed rotating balls cause the mechanical cracking of the C−C bonds and the subsequent exfoliation of the graphene sheets [26].

From a careful literature analysis, the available studies present very different experimental conditions and starting materials [27,28,29,30,31,32]. Therefore, a systematic study that can compare the effects of physical processing of carbon materials will allow a better understanding of the different contributions that influence the properties of carbon materials when subjected to mechanical treatment. Moreno−Fernández et al. [27] studied the effect of post−synthesis ball−milling of activated reduced graphene oxide on electrochemical performance, showing optimistic results with improved power density.

Although several types of electrolytes have been developed for supercapacitor applications and reported in the literature [30,31,32,33,34,35,36,37,38,39,40], high ionic concentration in the electrolyte is a prerequisite for capacitance improvement, and ionic liquids (ILs) present the required properties that make them suitable candidate [40,41,42,43,44]. Furthermore, ILs are less flammable when compared to common solvents, which contributes to improved safety issues and durability at high temperatures [45].

ILs are organic salts composed of organic cations and organic/inorganic anions, with high asymmetry and a solvent−free nature [46]. However, ILs present a high production and purification cost, which reduces their competitiveness when compared to traditional solvents. To overcome that situation, a solution can be found via the use of a new class of IL analogues known as deep eutectic solvents (DES), which present similar physical properties. DES are synthesized by complexing a quaternary ammonium salt with a variety of neutral, anionic and/or cationic species, namely metal chloride (type I), metal chloride hydrate (type II), a hydrogen bond donor (type III) or metal chloride hydrate complexed with a hydrogen bond donor (type IV) [47,48]. DES presents various advantages when compared to aqueous systems; better electrochemical stability, wider potential windows, and avoidance of gas evolution show that this new class of non−aqueous systems are the future of electrolytes for different applications [49,50,51].

In this work, commercial graphite was used as a starting material to prepare carbon nanomaterials through a dry ball milling processing. In this study, two different kinds of ball−milling equipment (Retsch Mixer Mill MM400 (Retsch GmbH, Haan, Germany) and IKA ULTRA−TURRAX^®^ (IKA®−Werke GmbH & Co., Staufen, Germany)) were used to modify commercial graphite powder. Commercial graphene and graphite were also used as reference materials to rate the structural and electrochemical performance of the modified graphite powders.

The dry ball−milling methods described in this work, may present a valuable route for obtaining nanoporous, high−specific surface area carbon materials through an innovative, easy, inexpensive, and environmentally−friendly route. Experimental data is interpreted based on surface interactions established between graphene/graphite−modified materials and type III DES electrolyte, composed of choline chloride and ethylene glycol as HBD, at a ratio of 1:2.

The obtained carbon nanomaterials will contribute to open the way to the development of easily obtainable electrodes for application in advanced electrochemical storage devices.

## 2. Materials and Methods

### 2.1. Chemicals and Solvents

Choline chloride (Sigma Aldrich, 99% (Merck KGaA, Darmstadt, Germany)) was dried in the oven overnight at 60 °C, before use; commercial graphene (platelets, 99.5 %, Iolitec Nanomaterials (IoLiTec−Ionic Liquids Technologies GmbH, Heilbronn, Germany)), commercial graphite (powder, Sigma Aldrich (Merck KGaA, Darmstadt, Germany)), ethylene glycol (EG) (Sigma Aldrich, 99+% (Merck KGaA, Darmstadt, Germany)), Nafion TM 117 (Sigma Aldrich (Merck KGaA, Darmstadt, Germany)), and N, N−Dimethylformamide (DMF; Sigma Aldrich, 99.8% (Merck KGaA, Darmstadt, Germany)) were used as received.

The eutectic mixture was formed by stirring the two selected components at 60 °C (proportions in Table 1) until a homogeneous, colorless liquid was formed. Prior to the electrochemical experiments, the solutions were de−aerated with nitrogen, and the cell was always kept under a nitrogen atmosphere.

### 2.2. Viscosity and Water Content of DES

The dynamic viscosity of the DES was measured using the automated Anton Paar DMA™ 4500 M microviscometer (Anton Paar GmbH, Graz, Austria), from 30 to 60 °C.

Water content (wt.%) was determined using a Karl Fischer titrator (831 KF Coulometer, Methrom (Herisau, Switzerland)) prior to the electrochemical studies. The sample solution was manually mixed to homogenize it before titrating, then 1 mL of sample was added to the dry methanol solvent (HYDRANAL™, max 0.01 wt.% water, Riedel−de−Haën (Honeywell Specialty Chemicals Seelze GmbH Charlotte, EUA)) and titrated with HYDRANAL™ Composite 5 Reagent (4.5–5.5 mg. mL^−^^1^ water equivalent, Riedel−de−Haën (Honeywell Specialty Chemicals Seelze GmbH Charlotte, EUA)) for moisture determination. Measurements were performed in triplicate.

### 2.3. Electrochemical Measurements

A three−electrode electrochemical cell consisting of a glassy carbon (GC) electrode (Methrom (Herisau, Switzerland), area 0.0721 cm^2^), a GC rod (Thermo Fisher (Kandel) GmbH, Germany) as counter−electrode, and a silver wire (Thermo Fisher (Kandel) GmbH, Germany) pseudo−reference electrode were used.

The preparation of the working electrode is described elsewhere [52,53]. Briefly, the working electrode was polished to a mirror−like finish before each experiment using 1 µm diamond suspension (Buehler, IL, USA) followed by 0.5 µm alumina powder (Buehler). The electrodes were rinsed with ultrapure water (purified through a Milli−Q Millipore system, Bioscience Research Reagents, CA, USA), followed by an electrochemical cleaning procedure using a sulphuric acid solution with a concentration of 0.5 mol. L^−^^1^ until a stable cyclic voltammogram profile was obtained. The electrode was dried under a nitrogen flow prior to all measurements.

The immobilization of the carbon material was preceded by the preparation of a dispersion of 5 mg of carbon in 950 µL DMF and 10 µL Nafion^®^ 117 mixture. Nafion was used as a binder to improve the adhesion of the material on the electrode surface. Binders such as Teflon, Nafion, and PVDF enhance the mechanical properties of films prepared from carbon particles, being typically used in supercapacitors, where they promote the performance stability over 5000 charge/discharge cycles [54]; however, they tend to introduce electric and ionic resistance to the electrode material. López−Chavéz [55] studied the effect of different binders, showing that is possible to use 1 wt.% Nafion solution as a binder with optimal electrode performance, reaching the maximum specific capacitance and minimum electric resistance contribution.

The dispersion went through ultrasonication for a given period (2 h) to obtain a homogeneous dispersion of the material. The amount of the carbon−coated on the GC electrode was obtained as an average of three measurements. The area density of the carbon material for the subsequent studies was calculated taking into consideration that the material and Nafion dispersion in DMF is homogeneous. The value of the area density was estimated to be ~6.63 × 10^−3^ g·cm^−^^2^.

To keep the dispersion subjected to the electrode’s active surface while avoiding the surrounding Teflon part, several small amounts of the suspension were dropped on the glassy carbon electrode surface using a micropipette. In the present work, 10 µL aliquots of the suspension were used for each of the five deposition stages. The electrode was left to dry at room temperature before use in the eutectic mixture.

The electrochemical measurements were performed using a computer−controlled AUTOLAB PSTAT 20 potentiostat/galvanostat from Eco Chemie, controlled by NOVA 2.1.5 software (Methrom (Herisau, Switzerland)). The experiments were performed at 30, 40, 50, and 60 °C. Voltammetric experiments were carried out at 50 mV·s^−^^1^, starting at 0 V towards the positive side.

Electrochemical impedance spectroscopy spectra (EIS, Methrom (Herisau, Switzerland)) were collected in the range of 20 kHz to 1 Hz, with frequencies logarithmically distributed and a sinusoidal signal of 10 mV (rms) superimposed over a dc potential. EIS measurements were made at 20 mV intervals. The differential capacitance was obtained from the EIS measurements, and the impedance data were fitted to an equivalent circuit using Nova 2.1.5. software version.

All the procedures followed to extract the capacitance from the impedance data are described in a work published by Silva et al. [56]. The EIS spectra were fitted to a simple R−CPE circuit, and the quality of the fitting was judged by the value of χ^2^ (<10^−3^), as can be seen in Figure 1. In the absence of Faradaic processes at the electrode surface, the fitting of an R−CPE circuit to the impedance data measured at electrode/deep eutectic solvent shows a nearly vertical straight−line slope.

The Nyquist complex plane impedance plot, Z″ (imaginary) versus Z′ (real), and Bode plot are presented in Figure 1 for G_REF/E200 at 0.60 V. The slight deviation observed on the Nyquist diagram to the real Z’ axis points to an almost ideal capacitive behavior.

Galvanostatic charge/discharge curves were collected at current densities of 1, 2, and 4 A·g^−^^1^. The specific capacitance in the three−electrode configuration was calculated from the galvanostatic discharge curves using Equation (1) as proposed by Stoller et al. [57].
(1)$$C=\frac{I\Delta t}{m\Delta V}$$
where I is the discharge current (A), Δt is the discharge time (s), ΔV is the potential window (V), and m is the weight of the carbon material in the electrode.

Although other methods for the calculation of the specific capacitance can be found in the literature (e.g. Vivekchand et al. [19]), we adopted Stoller et al.’s method [57], as the experimental conditions used in this work are closer to those described in the later publication. The method used by Vivekchand et al. [19] would give values systematically higher than those calculated using the method proposed by Stoller et al. [57].

According to Stoller et al. [57], the three−electrode cell can be used for the determination of the material’s electrochemical specific characteristics, and provides the best indication of an electrode material’s performance, while the two−electrode cell simulates the physical configuration, internal voltages, and charge transfer that occurs in a packaged supercapacitor. According to the same author [57], the cell should be cycled from 0 V to the maximum voltage (1 V in this case), because when a cell is first cycled or when it is cycled from a negative to a positive voltage there are increased current levels due to reversing the polarity of the cell.

### 2.4. Carbon Material Modification

The modification of the commercial graphite was performed by dry ball milling, using two different kinds of ball milling equipment, the IKA ULTRA−TURRAX^®^ Tube Drive (IKA®−Werke GmbH & Co., Staufen, Germany) (which allows grinding of a 15 mL volume sample, with 6 mm stainless steel balls 6000 rpm maximum rotation), which is lighter and cheaper, and the Retsch Mixer Mill MM400 vibromachine (Retsch GmbH, Haan, Germeny) (which uses a single 3 cm stainless steel ball bouncing at 25 Hz). Several treatments were made in the carbon materials; these experimental conditions are detailed in Table 2.

### 2.5. Carbon Material Characterization

Surface analysis was carried out using scanning electron microscopy, FEI Quanta 400 FEG/EDAX Genesis X4 M (FEI company, Hillsboro, OR, USA) at CEMUP. The image processing for the determination of the surface roughness (Ra) was performed using Gwyddion 2.53 software (Czech Metrology Institute, Okruzni, Czech Republic). Briefly, the SEM image was pre−processed. The preprocessing step included data leveling by mean plane subtraction, leveling of the data to make facets point upward, correcting lines by matching the height median, and correction of horizontal scars. The roughness parameters were evaluated using the roughness tool. The average roughness (Ra) was the result of the mean deviation of points from lines drawn across the sample image [58].

Atomic force microscopy (AFM) was performed using a Nano−Observer Atomic Force Microscope (PicoScan 2100, Les Ulis, France), operated in tapping mode and using SPM probes with a resonance frequency between 200 to 400 kHz. Image processing was performed with Gwyddion 2.53 software (Czech Metrology Institute, Okruzni, Czech Republic). The Ra parameter was obtained using the method presented above.

The surface area and pore parameters were determined using a nitrogen adsorption analyzer (TriStar Plus, Micromeritics, Norcross, GA, USA).

Attenuated total reflectance infrared measurements were performed using a Bruker FT−IR System Tensor 27 spectrophotometer (Systems Chemistry, Institute for Molecules and Materials, Heyendaalseweg, Netherlands) in the range of 4000 to 600 cm^−1^.

Raman spectra of the samples were recorded with a Raman spectrometer, Ramos PA532 Ostec (Moscow, Russia), using a 532 nm excitation wavelength. To perform the experiments, 200 mg of carbon material was previously dried overnight in the oven at 50 °C followed by at least 2 h under nitrogen flow.

## 3. Results and Discussion

### 3.1. Viscosity and Water Content of Ethaline 200

Water content is a parameter that can affect the physical properties of DES [47,48,49], namely the viscosity. Viscosity and water content are recognized as important characteristics of ionic solvents which influence their electrochemical performance.

Results for viscosity and water content are presented in Table 1 for temperatures between 30 °C and 60 °C. These data are in agreement with those of Salomé et al. [52], where a viscosity value of 16 cP was obtained at 75 °C.

No efforts were made to reduce the water content since we aimed in this work to use a straightforward procedure that would not require the use of a glove box to perform the electrochemical characterization of the carbon materials.

### 3.2. Characterization of the Carbon Materials

#### 3.2.1. Surface Area by Brunauer–Emmet–Teller Method

For a deeper understanding of the effect of milling on the morphology and specific area of the carbon materials, the specific surface area and micropore volume of commercial graphene, commercial graphite, and modified carbon materials were calculated from N_2_ adsorption−desorption isotherms using the Brunauer–Emmet–Teller (BET) method as presented in Table 3.

Graphene presents a BET specific surface area of 45.14 m^2^·g^−^^1^ and a pore volume of 0.0046 cm^3^·g^−^^1^ as measured by nitrogen adsorption at 77 K. On the other hand, graphite presents a BET specific surface area of 10.74 m^2^·g^−^^1^ and a much smaller pore volume of 0.00028 cm^3^·g^−^^1^.

Commercial graphene presents different values of specific surface area depending on the manufacturer; however, theoretical calculations predict a large surface area of single−layer graphene close to 2600 m^2^·g^−1^ [59]. Wasalathilake et al. [60] prepared graphene−based material displaying a specific area of 384.4 m^2^·g^−^^1^ with a pore volume of 0.73 cm^3^·g^−^^1^. The commercial graphene used in this study presents significantly lower values of specific surface area and pore volume, showing that the present commercial graphene sheets are less exposed.

The BET specific surface area determined for G_REF is approximately four times higher than that obtained for graphite powder, which is expected and in agreement with other experimental results, likely because graphite can suffer from overlapping and agglomeration of layers [61] as a consequence of the van der Waals interactions that can occur between the graphene sheets [53].

The effect on the surface area and pore volume of milling the graphite reference material for 10, 20, 40, and 60 min with the Retsch Mixer Mill MM400 equipment is also presented in Table 3. Increasing the ball milling time leads to an increase in the surface area and pore volume. For 10 min of milling, the increase is small (an almost two−fold increase in in the surface area and pore volume), while for 20 min milling time there is a marked increase in the surface area (20 times higher) and pore volume (80 times higher), reaching the maximum at 40 min milling time with a surface area of 308.6 m^2^·g^−1^ and a pore volume of 0.03163 cm^3^·g^−1^. Further increases in milling time result in a decrease of both surface area and pore−volume; nevertheless, the surface area (4×) and pore volume (15×) values are higher when compared to commercial graphite. A similar trend has been previously reported for milled graphite in work performed by Disma et al. [62] and Welham et al. [63]. Disma et al. [62] referred that the specific surface area of the ball−milling samples increased with the ball−milling time until a critical time was achieved, followed by a decrease. Following the same line of thought, Welham et al. [63] suggested that after the maximum surface area is reached, particles were rewelded, leading to a decrease in surface area (breakage/rewelding process). Chen. et al. [64] concluded that milling contamination did not affect the formation of this nanoporous structure. Similar results were reached by Zhang et al. [65] for ball−milled activated carbon, in which the specific surface area and pore volume diminished from 2137 m^2^·g^−1^ to 1683 m^2^·g^−1^, and 0.95 cm^3^·g^−1^ to 0.78 cm^3^·g^−1^, respectively, after 8 h of ball−milling. Mhadhbi et al. [66] used the discrete element method to simulate the correlation between several milling parameters (e.g., geometry, number, and size of balls and speed). The results showed that lower mass can improve milling performance, with the stainless−steel balls presenting a positive effect on milling efficiency as well as an increase in the milling velocity.

For further comparison, a parallel study was performed for graphite powder ball−milled with cheaper equipment, the IKA ULTRA−TURRAX^®^ Tube Drive ball miller.

The surface areas of the samples were determined using the BET procedure, with the obtained results presented in Table 3.

The products dry milled with stainless steel balls present an average surface area of 9.81, 10.80, and 11.33 m^2^·g^−^^1^ in samples obtained at different mill speed rotations of 2000, 4000, and 6000 rpm, respectively. G@2000 presented the highest pore volume, while G@6000 presented the highest surface area. Adsorption–desorption experiments established the existence of micropores with a pore volume approximately 29× smaller than those obtained by the Retsch Mixer MM400 milled samples. Although the Tube Drive ball miller can induce some transformation of the carbon material, this process lacks the energy to promote the structural changes required to form graphene sheets. In fact, for short rotations (2000 rpm) of Tube Drive ball milling, there is a reduction of surface area, probably due to the increase in particle agglomeration.

#### 3.2.2. Scanning Electron Microscopy Characterization

SEM imaging can contribute to the qualitative characterization of a material’s morphological structure (e.g., by allowing the comparison of the surface topography, composition, and the distribution of microscopic hollows in composites).

The morphology of the composite electrodes with G_REF and graphite samples were analyzed by SEM, and the images are represented in Appendix A, respectively.

The result of SEM G_REF at low magnification (5000× magnification) shows the existence of fewer stacks, with a smoother build−up morphology and larger inter−zone separation compared with the SEM image obtained for the graphite sample. The SEM images also show that the graphite has a more layered structure. The G_REF morphology presents a slight ripple and uniform surface compared with graphite, and shows randomly arranged aggregates, forming more irregular and closely spaced graphite microzones in the composite film (magnification 10,000×).

The SEM analysis agrees with the data collected from the BET analysis. The SEM images show distinct morphological differences between both composite electrodes (e.g., clear, and well−defined graphene sheets compared to smaller agglomerates obtained in the graphite sample), which may lead to a lower superficial area. The morphology of G10, G20, G40, and G60 was investigated via SEM (with representative micrographs presented in Figure 2), with magnifications of 5000× and 10,000×.

SEM micrographs from the surface analysis of the composite morphology with increasing time of the ball milling treatment show a noticeable change in the morphology of the carbon materials.

SEM images regarding the ball−milled graphitic samples using the MM400 equipment show that the range of particle size in the milled samples is wider compared with the graphene reference sample and the unmilled graphite powder. The SEM micrographs show particles with a flat shape (flake type) and with evidence for the presence of facets with treatments shorter than 40 min. The SEM image obtained for the G60 sample indicates that the particles tend to agglomerate, forming larger aggregates. This may be the reason for the lower specific area and pore volume obtained for this sample. By increasing the magnification of the image (which allows a closer examination of the large aggregates that present some faceting), however, the majority have a rounded shape, representing the possibility of the loss of edges due to abrasion during the milling process. Consequently, 40 min of milling seems to lead to a size reduction by fracturing the larger particles into smooth and smaller ones; however, when the optimal milling time is exceeded, it seems that the smaller particles themselves become agglomerated (approx. 5–10 µm). The fracturing procedure of large clusters is no longer in equilibrium with the agglomeration trend of small particles; subsequently, the surface area begins to drop [64].

The graphite dry ball−milling procedure seems to promote the generation of a new surface that may be accompanied by the increase in surface area concomitant with a reduction in particle size, signifying that no large clusters can be formed [67]. The arrangement of these large clusters will influence the complex surface area trend, as supported by BET analysis during the additional milling period.

The SEM and BET data strongly suggest that many micropores were created in the first stage of the milled graphite sample by the ball milling tools, resulting in particle fracture induced by ball impacts. The subsequent reduction reported in BET surface area analysis is associated with agglomeration effects (larger clusters; [64]), which are evident in the G60 SEM micrograph (Figure 2).

The SEM micrographs of IKA ULTRA−TURRAX^®^ material (Appendix A) ball−milled samples reveal an insignificant modification in the sample morphology after ball milling compared with the starting material’s flake−like structure. The impact forces from ball collisions seem not to cause relevant mechanical stress in graphite structure compared with the Retsch Mixer Mill. Despite the SEM images obtained for the G@2000 sample, it seems to be more uniform, probably caused by the sliding of the graphite layers. The IKA ULTRA−TURRAX^®^ ball milling procedure did not effectively thin or break the graphite particle down further, presenting a specific area lower than the graphene reference material, and similar values to the starting graphite material.

Surface roughness is also a parameter that needs to be considered more accurately to enhance electrical properties. Several authors have shown that capacitance increases with surface roughness [68,69,70]. The surface roughness determination was performed through the analysis of the SEM images. The surface roughness (R_a_) for the different carbon materials is presented in Figure 3, with the associated standard deviation for each sample.

These results show that surface roughness is strictly associated with the variation of S_BET_, showing that the increase of S_BET_ leads to an increase in the surface roughness of the carbon material, with the G40 sample presenting a maximum surface roughness of 75 nm, against 49 nm for the commercial graphite.

AFM analysis of the topography of the ball−milled samples (G10 to G60) was also performed (Appendix A) to complement the SEM studies, and the R_a_ parameter (Appendix A) was also determined, following the same method presented above. The topography at both magnifications (50 µm × 50 µm and 10 µm × 10 µm) shows that there is a uniform dispersion of the carbon materials on the GC electrode’s surface. There is a visible increase in grain size in samples G20 and G40, where the R_a_ obtained through AFM analysis presents the same tendency as presented in Figure 3, with G40 presenting a maximum surface roughness of 189 nm. The discrepancy in the obtained values is due to the different characterization techniques.

#### 3.2.3. ATR−IR and Raman Spectroscopy Characterization

ATR−IR and Raman spectroscopy methods were also used for the characterization of the samples to study the evolution of the vibrational spectrum of the G_REF and modified samples due to the ball−milling process.

Figure 4 shows the ATR−IR spectra of G_REF, graphite, and ball−milled graphite samples. All samples present very similar peaks with different intensities.

G_REF and graphite ATR−FTIR analysis is shown in Figure 4a, presenting three prominent peaks (1922 cm^−^^1^ and 2112 cm^−^^1^, coincident for both materials, and at 2330 cm^−^^1^ and 2331 cm^−^^1^ for graphite and G_REF, respectively). These peaks are related to the C=O bond stretching vibration from the gaseous CO_2_ presented in the atmosphere and absorbed on the walls of the porous carbon structure [71]. The peak intensity related to the C=O bond is determined by the concentration of the impurities and the thickness of the carbon structure [72], with both samples presenting the same peak intensity.

Three peaks appear between 3000–4000 cm^−^^1^ (3631 cm^−^^1^, 3745 cm^−^^1^, and 3850 cm^−^^1^) for both G_REF and commercial graphite, suggesting the possibility of adsorbed water molecules being due to the −OH stretching fundamental vibration **v**_3_ and **v**_1_ in the water monomer [73,74,75], with a higher intensity in commercial graphite. In G_REF, a low−intensity peak appears at ~3000 cm^−^^1^, related to the C−H sp^2^ bond from the graphitic aromatic ring [71]. Huang et al. [76] and Ciplak et al. [77] performed ATR−IR analysis of graphene, presenting similar peaks.

Figure 4b shows the ATR−IR analysis for the ball−milled graphite samples through the MM400 equipment. All peaks previously stated for G_REF and graphite are present in all MM400 ball−milled samples, with higher peak intensity when compared to the commercial graphite and G_REF samples. The G40 sample presents the peaks with higher intensity, which can be associated with the sample with higher surface area, in which the peak intensity is well correlated with the surface area obtained from the BET method, indicating a higher CO_2_ and water absorption due to the higher exposed area of the carbon material. G40 and G10 transformed the three peaks between 3000 and 4000 cm^−^^1^ into a single peak very close to 3746 cm^−^^1^. The same behavior is presented in Figure 4c for the Ultra−Turrax ball−milled samples, with G@2000 presenting a single peak at 3000–4000 cm^−1^ compared with the three peaks presented by the other samples. This might be related to the broadening of the peaks associated with higher –OH bonding interaction, creating higher disorder due to the higher inhomogeneity of the samples [78]. Figure 4b,c present a peak around 1600 cm^−1^ for the ball−milled samples for the O−H groups from water molecules [71], presenting the same tendency regarding the three peaks appearing between 3000–4000 cm^−1^.

To confirm the ATR−IR analysis, Raman spectroscopy was used for the characterization of the studied carbon materials to focus on the characteristic peaks associated with graphitic−based materials. Raman spectra related to the first order Raman region of G_REF and commercial graphite is presented in Figure 5a, followed by the modified graphite samples through MM400 ball−milling (Figure 5b) and Ultra−Turrax ball−milling (Figure 5c).

The Raman spectra of all the studied samples are presented in Appendix A, alongside the deconvolution study for the D and G bands. In this study, special attention has been given to the Raman region between 1000 and 2000 cm^−^^1^. The Raman region higher than 2000 cm^−^^1^ includes the 2D Raman mode, being the strongest Raman mode in single−layer graphene [79], and two D band phonons with opposite, non−zero momentum. In the studied samples, the intensity of the 2D band (~2900 cm^−^^1^ [79]) is insignificant, which indicates that we are dealing with multilayer graphene (I_2D_/I_G_ < 0.5 [80]). Therefore, the deconvolution of the Raman bands was performed only for the D and G bands.

As expected for carbon−based materials, all studied samples reveal the D (~1350 cm^−1^) and G (~1600 cm^−1^) bands [81,82,83].

For all the Raman spectra of the studied graphitic materials, the first order region was deconvoluted into five peaks (D4, D1, D3, G, and D2) using the Gauss function (Appendix A) [84,85,86,87,88].

The G band is a sharp band appearing around 1600 cm−1 in the spectrum of graphene, and it is associated with single−crystal graphite [84,89,90]. The G band originates from the stretching of the C−C bond in the hexagonal plane and is common to all sp2 carbon systems [91]. Appendix A, summarizes the main results regarding the Raman spectra of the different studied carbon materials.

The D band is known as the disorder or defect band, representing a ring breathing mode from sp^2^ carbon rings [82]. The intensity of the D band is directly proportional to the level of defects in the sample [89]. For all the G_REF, commercial graphite, and ball−milled graphitic samples, the D band is around 1350 cm^−1^.

Figure 5 reveals the increase in the intensity of the D and G bands with the increase in ball−milling time for the MM400 samples (G10, G20, G40, and G60), except for the G60 sample, which presents a decrease in intensity. The same tendency is not observed for the Ultra−Turrax ball−milling samples, showing inconsistent behavior.

The degree of disorder and the average size of the in−plane sp^2^ domains are determined from the intensity ratio of the D and G bands (I_D_/I_G_) [90]. The I_D_/I_G_ ratio for the different samples is presented in Appendix A, showing that the ratio I_D_/I_G_ ratio of G_REF is almost half compared to graphite (0.15 and 0.26, respectively), indicating that the G_REF sample presents far fewer defects in its carbon structure. Taking into consideration the MM400 ball−milling samples, the I_D_/I_G_ ratio increases with the increase in ball−milling time, decreasing only for the G60 sample, thus showing that the increased level of defects is strictly associated with the increase in specific surface area, suggesting the formation of new domains of conjugated carbon atoms [92]. For the samples related to the Ultra−Turrax ball−milling, the same tendency is not verified, presenting an anomalous behavior regarding the I_D_/I_G_ ratio variation with changes in speed rotation.

The commercial graphite sample presents a similar I_D_/I_G_ ratio compared to that obtained by Labunov et al. (0.24) [93]; however, Dubale et al. [94] presented an increased value in the I_D_/I_G_ ratio of 1.31, indicating that the studied graphite in this work presents fewer defects. The same authors presented the I_D_/I_G_ ratio of 0.403 for graphene, which is significantly higher than the graphene studied in this work.

### 3.3. Electrochemical Behavior of G_REF and Commercial Graphite Modified Electrodes

To establish reference values for the electrochemical performance of the different carbon materials, a systematic study was carried out using a three−electrode electrochemical cell and a working electrode modified with commercial graphene (G_REF) and commercial graphite thin films; the experimental characterization of the electrochemical properties of these systems are displayed in Figure 6 and Figure 7, respectively.

Figure 6 and Figure 7a display the galvanostatic charge–discharge curves recorded with current densities 1, 2, and 4 A·g^−^^1^. At 1 A·g^−^^1^ the capacitance values obtained were 5.45 F·g^−^^1^ for G_REF/E200 and 4.27 F·g^−^^1^ for the commercial graphite, as displayed in Table 3.

From the analysis of Figure 6 and Figure 7b (scan rate evaluation at 30 °C) and Figure 6 and Figure 7c (temperature effect), the G_REF system presents a rectangular shape in the cyclic voltammetric profile, showing almost ideal capacitance behavior, while the graphite−modified GC electrode does not follow that pattern. The scan rate effect in both carbon materials presents an increase in both anodic and cathodic current with the increase in the scan rate.

The temperature effect was studied for both G_REF and graphite immersed in E200, and the detailed results are presented in Figure 6 and Figure 7c–f.

The gravimetric capacitance−potential curves (Figure 6 and Figure 7c) for both G_REF and the graphite materials in E200 present a U−shape. Costa et al. [95] studied the electrochemical interfaces of DES−based on choline chloride mixtures with ethylene glycol, 1,2−propanediol, and urea at the Hg electrode, presenting similar U shapes for the studied DES. In this study, the material of the modified electrode does not present any effect, since it does not change the shape of the curves in terms of either the anodic or cathodic potentials.

Both G_REF/E200 and graphite/E200 show an increase in gravimetric capacitance with increasing temperature (Figure 6 and Figure 7a). Even though there is a substantial drop in electrochemical performance in the initial cycles, the retention in capacitance starts to stabilize after 400 cycles for G_REF/E200 (Figure 6), and the stabilization does not occur for graphite/E200 for 1000 cycles. There is an inconsistency in the variation of the capacitance retention with temperature; however, it is possible to assume that the tendency is to decrease the capacitance retention by increasing temperature. This might be due to the deterioration of the carbon composite film present on the electrode’s surface.

### 3.4. Electrochemical Behavior of Ball−Milled Graphite Modified Electrodes

Boosting the capacitance by increasing the interfacial area per volume has been the focus of scientific research for more efficient energy storage technologies aiming to increase supercapacitor energy [96].

Figure 8 illustrates the experimental results gathered in the electrochemical study regarding the effect of dry milling the graphite reference material for 10, 20, 40 and 60 min with the Retsch Mixer Mill MM400 equipment, compared to the G_REF sample at 30 °C. The ball−milling time effect was evaluated by cyclic voltammetry (Figure 8a), gravimetric potential curves (Figure 8b), galvanostatic charge–discharge curve profiles (Figure 8c), discharge gravimetric capacitance (Figure 8d), and capacitance retention (Figure 8e). The temperature effect on capacitance is presented in Appendix A for all ball−milled carbons for temperatures between 30 °C and 60 °C.

Cyclic voltammograms (CV) obtained between 0 and 1 V measured at different interfacial composite surfaces immersed in E200 are shown in Figure 8a. The CV curves display the characteristic capacitor−like profile in the shape of almost rectangular cyclic behavior for the graphite prepared for 20 and 40 min. The CV comparison also shows an abrupt decrease in current gravimetric density for the material prepared, using a longer interval (60 min) for the ball milling process. This may result from a decrease in the penetration ability of the electrolyte ions to reach the inner spaces of pores induced by the applied potential. This behavior may also reflect the relationship established between the samples’ milling time procedure and the ability of ions to more easily reach the pores under conditions of finely controlled microstructure.

The CVs do not show evidence of any pseudocapacitance (faradaic process) contribution, which may indicate that the gravimetric capacitance determined in this work is purely capacitive. The gravimetric capacitance (Figure 8b) increases with the increasing time of sample treatment, presenting higher values compared to G_REF for the G20 and G40 samples. These results agree with the changes reported in the BET study, indicating that the increase in both pore volume and specific surface area leads to an increase in gravimetric capacitance. The G40 ball−milled graphite with an approximate surface area of 308.6 m^2^·g^−^^1^ presented the highest specific capacitance, with an approximate value of 25 F·g^−^^1^, four times greater than the commercial graphene sample (G_REF). An increase in surface area and pore volume leads to more surface sites that are available for charge storage. Several authors have studied the effects of surface area and pore volume on the specific capacitance of EDLCs, showing that there is an increase in capacitance with an increase of both parameters [97,98,99]. Eguchi et al. [100] studied the ball−milling effect on high−specific surface area activated carbon (>3000 m^2^·g^−^^1^) manufactured from petroleum coke employing KOH activation with different ball−milling times, showing that prolonged milling led to a degeneration of pores and a decrease in both gravimetric specific capacitance and pore volume.

It is known that only the pore surface accessible by the ions can contribute to double−layer capacitance. The bigger the ions present in the electrolyte, the larger the pore size required for the carbon materials [101]. The benefits related to a high surface area need to be balanced by the associated loss of stability, in which a careful compromise needs to be taken into consideration. In this case, using DES as an electrolyte, pore size and surface area need to present higher values when compared to aqueous electrolytes.

This effect is reversed with the G60 sample, where the increase in milling time led to a substantial decrease in capacitance. A decrease in capacitance can be explained by the overlapping and agglomeration of graphite layers [61]. This phenomenon may be due to the van der Waals interactions that can occur between the graphene sheets [53].

A galvanostatic charge−discharge test was performed on the different graphitic materials/ethaline 200 interfaces to evaluate its capacitive behavior; the results obtained at current density 1 A·g^−^^1^ are displayed in Figure 8c. For the G_REF and ball−milled samples (G10 and G60), the response of the interface approaches the ideal linear charge–potential relationship. For the G20 and G40 samples, the total charged interface initially drops, followed by a constant capacitive performance. A slight IR drop in the galvanostatic charge–discharge curves is more evident at the G20 and G40/ethaline 200 interfaces. Since IR drop is a direct measure of electrolyte resistance, influences the overall power performance, and has a significant impact on electrochemical measurements, these effects must be taken into consideration throughout the interpretation of electrochemical data.

The cycle rate performance of the specific capacitance was also evaluated from the charge–discharge cycles for 1000 cycles at a current density of 1 A·g^−^^1^ and is represented in Figure 8d.

These results point to excellent stability and reliability at these current charge/discharge cycles. After 1000 cycles, capacitance retention was as high as 80% for the G20 and G40 samples (Figure 8e). The high specific capacitance (25.10 F·g^−^^1^) also results from the high pore volume (0.03 cm^3^·g^−^^1^) and high surface area (308.58 F·g^−^^1^) of the G40 sample. Schutjajew et al. [102] recently studied the effects of the pore size and specific surface area of carbon materials in an ionic liquid electrolyte through cyclic voltammetry for supercapacitor applications. A clear correlation was found, showing that higher pore volume leads to higher peak areas and currents, with a specific surface area of 881 m^2^·g^−^^1^ and pore volume of 0.123 cm^3^·g^−^^1^ for the highest specific capacitance of 151.8 F·g^−^^1^ at a current density of 0.1 A·g^−^^1^.

For further comparison, a parallel study was performed for graphite powder ball−milled with cheaper equipment, the IKA ULTRA−TURRAX^®^ Tube Drive ball miller; both morphological (SEM) and electrochemical studies are presented in Appendix A. Other studies changing the milling time (30, 40, and 50 min) for sample G@6000 were performed, without observing significant changes (Appendix A). Thus, results for the samples G@2000 to G@6000 with 1h of milling were presented.

The electrochemical studies grafting the glassy carbon electrode with G@2000, G@3000, and G@6000 materials (Appendix A) show far less specific capacitance compared to the previously−reported ball−milled carbon materials (G10 to G40). Even commercial graphite electrochemically outperformed the material obtained by the ULTRA−TURRAX^®^ method. This electrochemical performance is in line with the information obtained for the morphological data and shows that the ULTRA−TURRAX^®^ method does not allow the required changes in the graphite, while longer times could not be used due to overheating during the process causing the plastic tube to rupture.

Both ball−milling methods present lower gravimetric capacitance compared with other studies performed using aqueous electrolytes. Zdolsek et al. [103] presented carbon materials synthesized by ionic liquids through hydrothermal carbonization, showing performances around 148 F·g^−^^1^ at 1 A·g^−^^1^, where the highest value obtained in this work is 25.17 F·g^−^^1^. The same happens in Iakunkov’s work, in which the workgroup prepared rGO materials through ball milling and activation temperatures with a capacitance of 174 F·g^−^^1^ [104]. However, these results were obtained using two−electrode coin cells with higher electrode areas (around 1 cm^2^), and therefore it is not valid to make a direct comparison of the experimental data.

## 4. Conclusions

Commercial graphite was subjected to two dry ball milling methods, with the aim of evaluating the contribution of the physical treatment to the properties of the carbon materials. Assessment of the effectiveness of the dry ball−milling process was made through the capacitance measurement of glassy carbon composite electrodes, using E200 as an electrolyte. Further, the structural changes of the ball−milled and composite powders were characterized using BET and SEM analysis. Raman and ATR−FTIR were performed, demonstrating the changes in the characteristics of the different modified samples. Distinguishing features were obtained between the different ball−milled samples and commercial materials.

The optimal ball milling time was achieved at 40 min with the Retsch Mixer Mill MM400. G40 ball−milled graphite material presented the highest surface area (308.6 m^2^·g^−1^), pore volume (0.03163 cm^3^·g^−1^), and specific capacitance (25 F·g^−1^), four times higher than G_REF. The results obtained in the present study indicate that the increase in gravimetric capacitance is strongly related to the increase in porosity and surface area. For longer milling periods (G60 sample), the surface area decreased, as well as the gravimetric capacitance. The use of ULTRA−TURRAX^®^ Tube Drive did not have a significant impact on the carbon material structure; 1 h of treatment at 4000 rpm was equivalent to 10 min of treatment using the MM400 vibromachine. These results allowed to optimize the electrode capacitance by increasing the interfacial area per volume of carbon materials through an easy and sustainable method for further scaling up to a coin cell setup.

These results highlight the potential of using ball−milled graphite materials to develop a new generation of sustainable and environmentally−friendly carbon−based materials, with potential applications in advanced storage devices such as high−energy supercapacitors using deep eutectic solvents.

## Figures and Tables

**Figure 1 nanomaterials-11-03258-f001:**
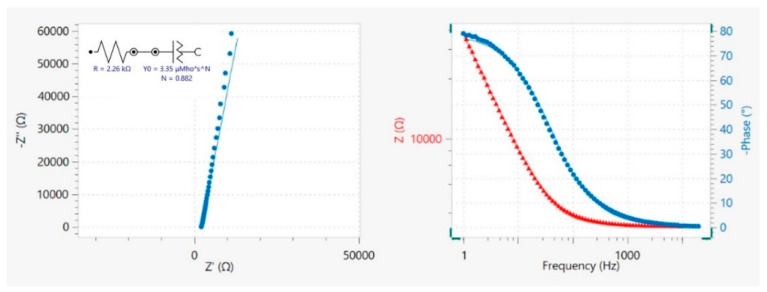
Nyquist diagram (−Z” vs. Z’) and Bode plot (|Z| and −φ vs. frequency) measured at the G_REF/E200, at 0.60 V, where experimental data (dots) and the line represent the fitting using an equivalent circuit R−CPE. Figure obtained from the NOVA 2.1.5 software.

**Figure 2 nanomaterials-11-03258-f002:**
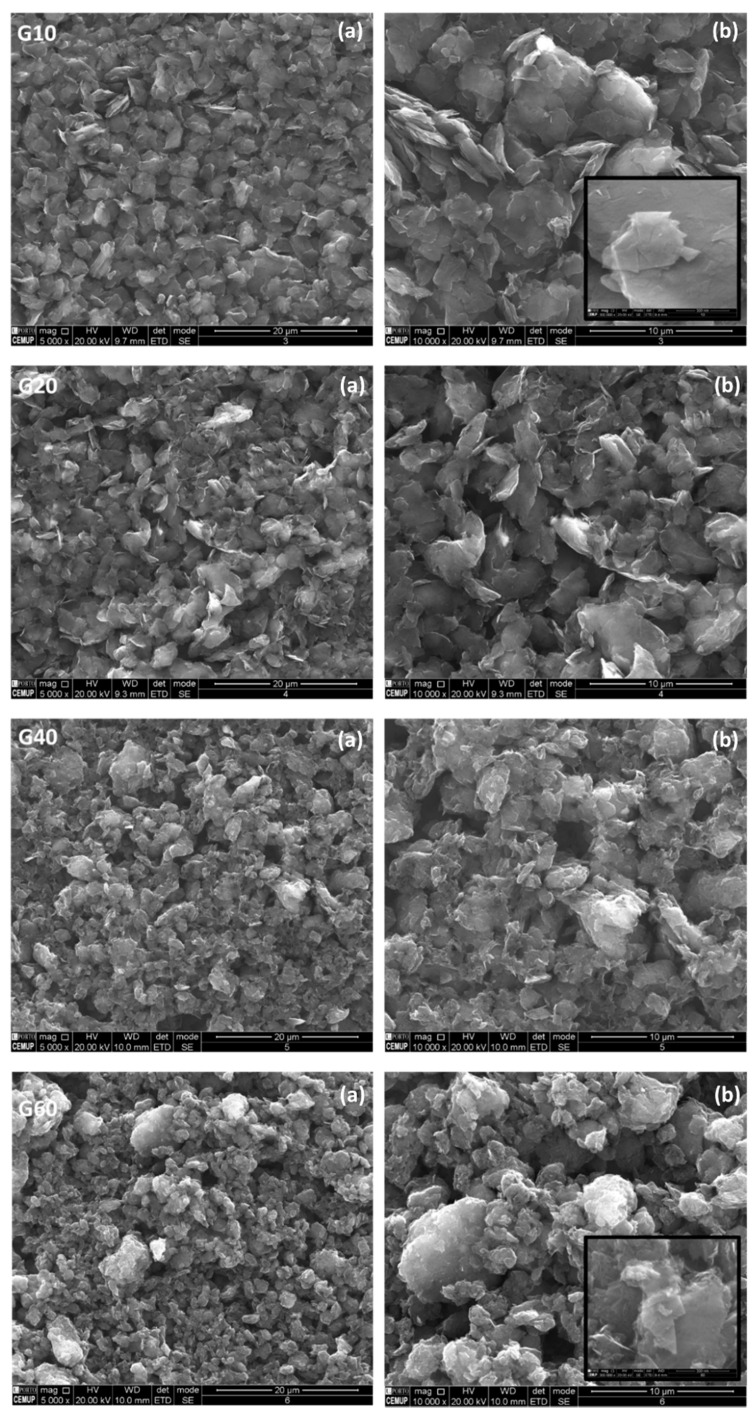
Electron microscopy images showing the structure of G10, G20, G40, and G60 samples with 5000× (**a**) and 10,000× (**b**) magnification. Inset images in Samples G10 and G60 present a 300,000× magnification.

**Figure 3 nanomaterials-11-03258-f003:**
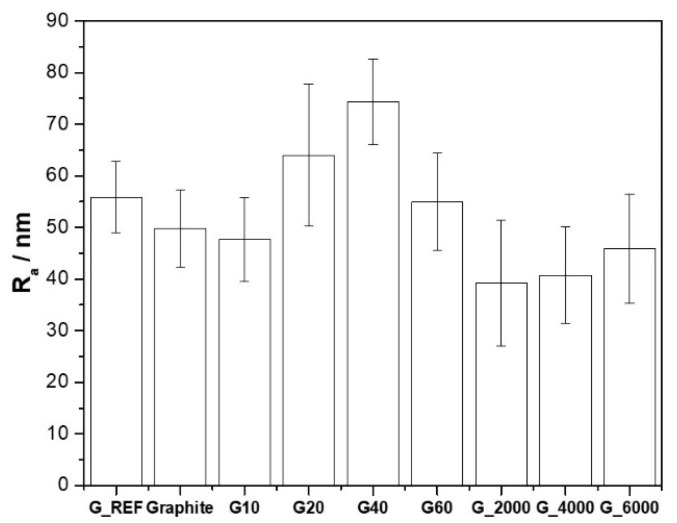
Surface roughness (R_a_) for the different carbon materials.

**Figure 4 nanomaterials-11-03258-f004:**
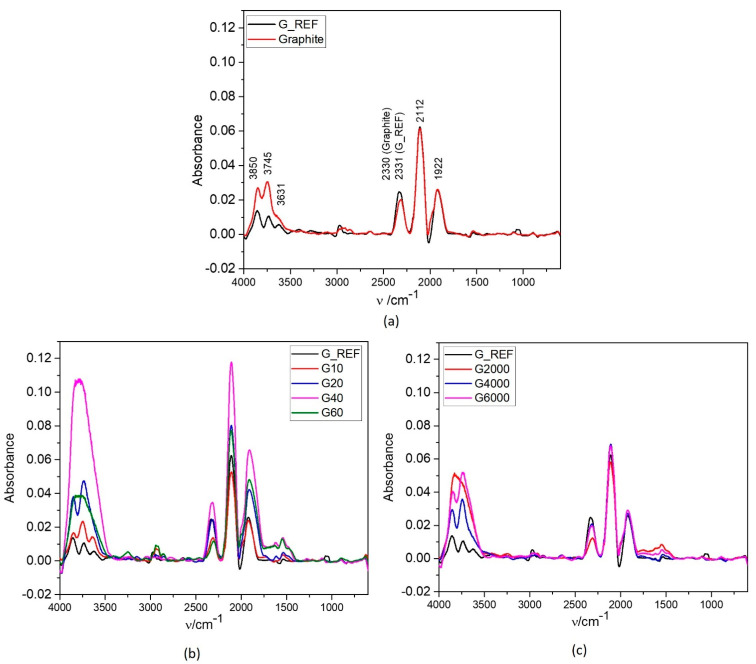
ATR−IR spectrum of commercial graphene (G_REF), commercial graphite (**a**), and modified graphene samples through the two different ball milling processes (**b**,**c**).

**Figure 5 nanomaterials-11-03258-f005:**
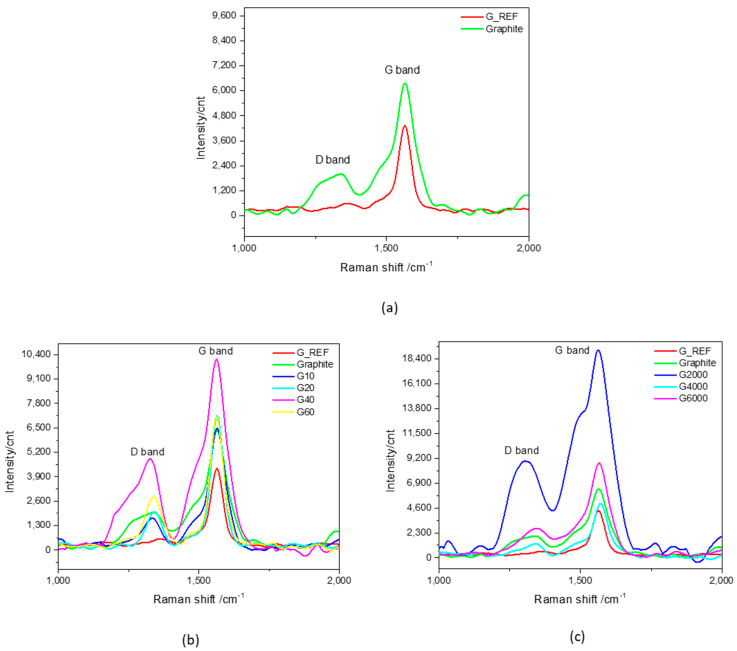
Raman spectra (1000–2000 cm^−1^) of (**a**) G_REF and commercial graphite, (**b**) MM400 ball−milled graphitic samples, and (**c**) Ultra−Turrax ball−milled graphitic samples.

**Figure 6 nanomaterials-11-03258-f006:**
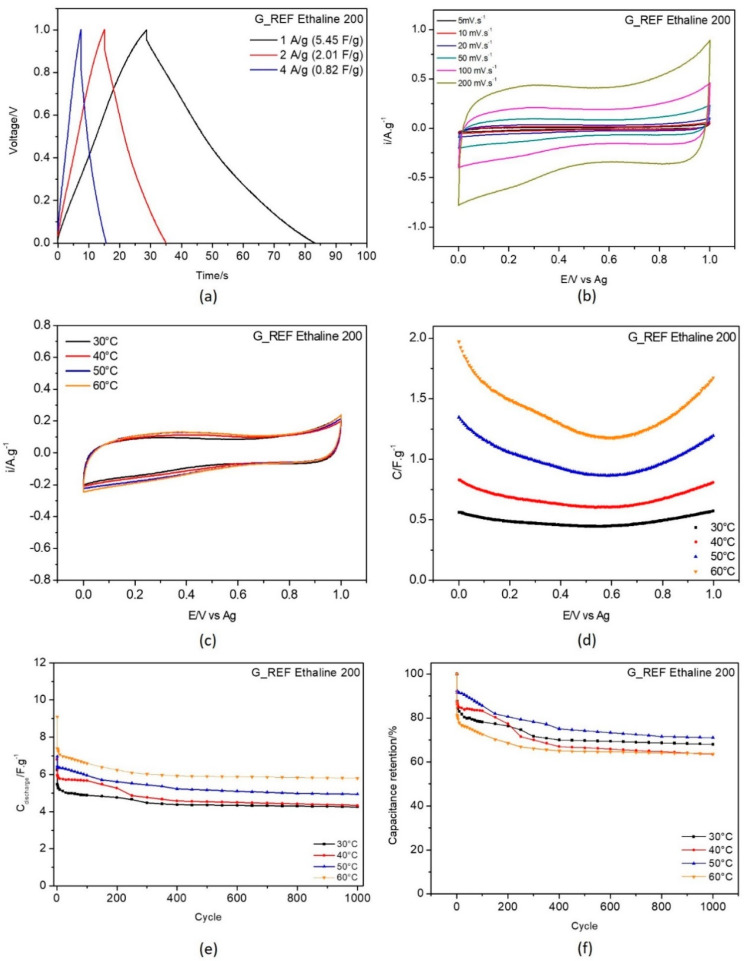
Electrochemical testing of G_REF in E200 electrolyte: (**a**) galvanostatic charge−discharge curves recorded with current density 1, 2 and 4 A·g^−1^ at 30 °C; (**b**) CV curves recorded at scan rates 5, 10, 20, 50, 100 and 200 mV·s^−1^ at 30 °C, temperature effect on (**c**) cyclic voltammetry at 50 mV.s^−1^; (**d**) capacitance−potential curve; (**e**) discharge gravimetric capacitance for 1000 cycles; and (**f**) capacitance retention.

**Figure 7 nanomaterials-11-03258-f007:**
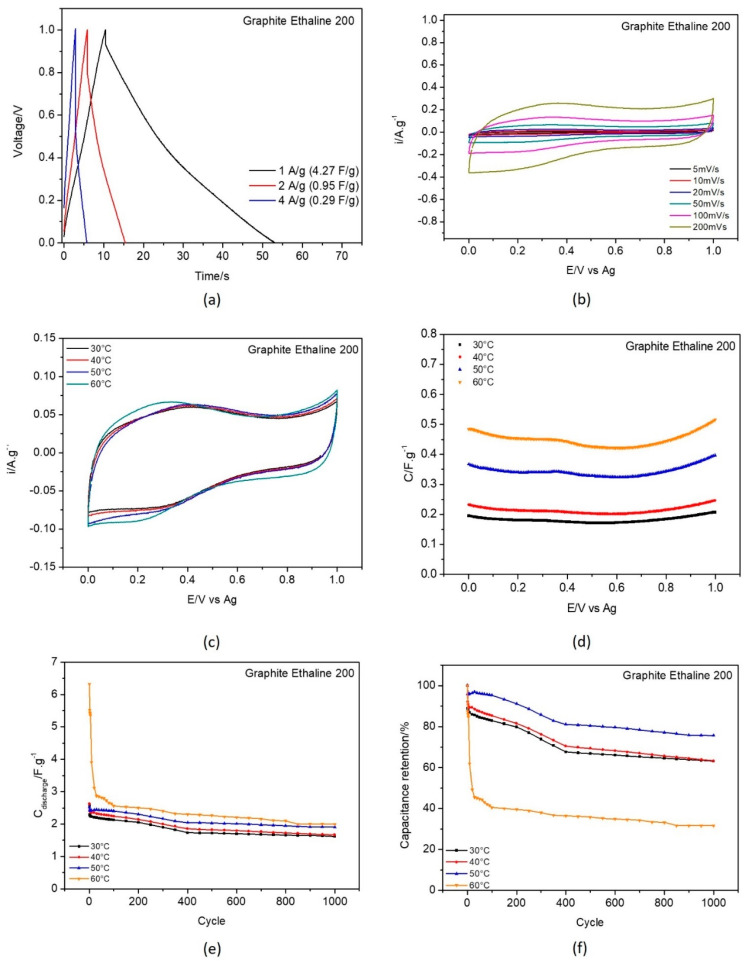
Electrochemical testing of commercial graphite in E200 electrolyte: (**a**) galvanostatic charge−discharge curves recorded with current density 1, 2 and 4 A·g^−1^ at 30 °C; (**b**) CV curves recorded at scan rates 5, 10, 20, 50, 100 and 200 mV.s^−1^ at 30 °C, temperature effect on (**c**) cyclic voltammetry at 50 mV.s^−1^; (**d**) capacitance−potential curve; (**e**) discharge gravimetric capacitance for 1000 cycles; and (**f**) capacitance retention.

**Figure 8 nanomaterials-11-03258-f008:**
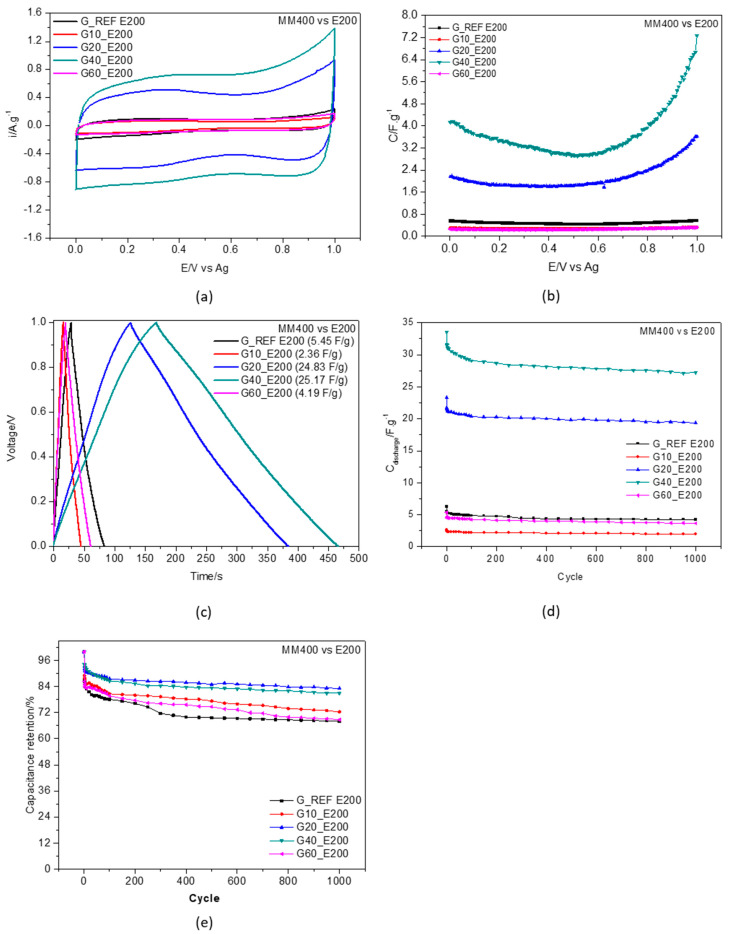
Ball milling effect using Retsch Mixer Mill MM400 for G_REF, G10, G20, G40, and G60 at 30 °C in E200. (**a**) cyclic voltammetry at scan rate of 50 mV.s^−1^ at 30 °C; (**b**) capacitance–potential curve; (**c**) galvanostatic charge–discharge curves recorded with current density 1 A·g^−1^; (**d**) discharge gravimetric capacitance for 1000 cycles; (**e**) capacitance retention (at 1 A·g^−1^).

**Table 1 nanomaterials-11-03258-t001:** Composition and commercial designation of the DES used, with water content and viscosity measurements.

DES	Composition	Molar Ratio	Choline Chloride	HBD	Water Content (wt.%) **	Viscosity (Cp) ***
ethaline 200 * (E200)	Choline chloride (ChCl) + ethylene glycol (EG)	1 (ChCl):2 EG	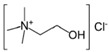	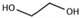	8.8 ± 0.5	30 °C: 64.3 ± 1.340 °C: 48.7 ± 1.850 °C: 35.1 ± 0.960 °C: 25.3 ± 0.3

* Trade name, ** measured at room temperature, right after the DES preparation *** measured between 30–60 °C, right after the DES preparation.

**Table 2 nanomaterials-11-03258-t002:** Treatment for carbon modifications and respective terminology.

Carbon Material	Terminology	Code	Milling	Treatment
Graphene *	Commercial graphene (11–15 nm, 99.5%, Iolitec nanomaterials)	G_REF	−	As received
Graphite	Commercial graphite (<20 µm, synthetic, Sigma Aldrich)	Graphite	−	As received
	2000 rpm	G@2000	ULTRA−TURRAX^®^	Ball milling for 1 h changing the rotation speed
	4000 rpm	G@4000		
	6000 rpm	G@6000		
	10 min. MM400	G10	Retsch MM400	Ball milling at 25 Hz changing the milling time
	20 min. MM400	G20		
	40 min. MM400	G40		
	60 min. MM400	G60		

* Reference graphene sample.

**Table 3 nanomaterials-11-03258-t003:** Surface area, pore volume, and specific capacitance determined for commercial graphene (G_REF), commercial graphite, and commercial graphite subjected to milling using a Retsch Mixer Mill MM400 (samples G10, G20, G40 and G60) and IKA ULTRA−TURRAX^®^ Tube Drive (samples G@2000, G@4000 and G@6000). Specific capacitance was evaluated at the interface between the composite graphite/E200 with current density 1 A·g^−1^, at 30 °C.

Samples	S_BET_ (m^2^·g^−1^)	Pore Volume (cm^3^·g^−1^)	Specific Capacitance (F·g^−1^)
G_REF	45.14	0.00461	5.45 ± 0.96
Graphite	10.74	0.00028	4.27 ± 0.85
G10	18.15	0.00050	2.36 ± 0.21
G20	235.80	0.02227	24.83 ± 2.33
G40	308.58	0.03163	25.10 ± 2.22
G60	41.84	0.00419	4.19 ± 1.01
G@2000	9.81	0.00103	1.95 ± 0.35
G@4000	10.80	0.00081	2.57 ± 0.44
G@6000	11.33	0.00088	2.59 ±0.31

## Data Availability

Not applicable.

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
