# Peer review of "Sustainable Preparation of Nanoporous Carbons via Dry Ball Milling: Electrochemical Studies Using Nanocarbon Composite Electrodes and a Deep Eutectic Solvent as Electrolyte"

_nanomaterials, 2021, doi:10.3390/nano11123258_

Round 1

Reviewer 1 Report

This work produced nanoporous carbon materials from graphite via dry-ball-milling procedures and studied the effect of the milling conditions on the properties of modified carbons. In addition, deep eutectic solvent was used as electrolyte for electrochemical performance study.

  1. For graphite from Retsch MM400, 60 min is too long and results in the lowest surface area. However, all samples processed via ULTRA-TURRAX were conducted for 1h at different speeds. Is 1h also too long for ULTRA-TURRAX processed graphite to achieve optimal surface area?
  2. DES electrolyte has larger voltage window than that of aqueous electrolyte. However, in this manuscript, the authors only demonstrate a 1 V voltage window in their electrochemical performance measurement? Please explain.
  3. From SEM images (Fig. 2), the difference of particle sizes among G10 to G60 can be distinguished. However, the surface morphology is not very clear. The authors should provide SEM images with higher resolution.
  4. About the surface roughness of carbon materials, please provide more detailed description on the calculation method.
  5. What’s the scan rate in Fig. 8a?
  6. In Fig. 8b, G20 has higher capacitance than G40, which contradicts with other result. Please check.

Author Response

The authors acknowledge the constructive comments raised by the reviewer and would like to thank you for the opportunity to improve the proposed manuscript. Several changes have been made in the manuscript in order to accommodate the relevant questions raised by the reviewer. Please find attached the file with the answers to your questions and suggestions.

Reviewer 2 Report

In this paper, the authors introduced, commercial graphite was used as starting materials to prepare carbon nanomaterials through dry ball milling processing. In this study, two different ball-milling equipment (Retsch Mixer Mill MM400 and IKA ULTRA-TURRAX®) were used to modify commercial graphite powder. Commercial graphene and graphite were also used as reference materials to rate the structural and electrochemical performance of the modified graphite powders. The idea behind this is interesting. However, I still have quite a number of concerns in this manuscript. There are times where there are not enough data to support the conclusions of the author. Please see some of the major concerns below.

1.The information for the Composition and commercial designation of the DES used, and water content and viscosity measurements and Nyquist diagram (-Z” vs. Z’)  figures is not enough. The authors should give much more information about this. For example in figure 2 the gray color can be erase for better obtaining better clarity, also what are the blue dot points meaning?

  1. The authors should give much more information about the novelty of this paper, especially the effect of using this commercial graphite, which applications can be used this device?

  1. The tolerance analysis, which can offer a good guide for the fabrication or using commercial graphite requirement, and the key parameters, need to be added in the results section.

  1. More references need to be included in the introduction part to understand the applications of using Raman spectroscopy for device applications.

  1. Super-resolved Raman spectroscopy,- Spectroscopy Letters, 2013
  2. Improving Raman spectra of pure silicon using super-resolved method

- Journal of Optics, 2019

  1. Super-resolved Raman spectra of toluene and toluene–chlorobenzene mixture

- Spectroscopy Letters, 2015

  1. Much more discussion about the results should be given in this paper, especially the author needs to provide enough physicals mechanism analysis about the results.

Author Response

(The authors gave the same response as above.)

Round 2

Reviewer 1 Report

The authors have addressed my concerns. Thus I recommend its publication.

Reviewer 2 Report

The new version can be published